# Self-consistent hardness measurements spanning eleven decades of strain rate on a single material surface

**Luciano Borasi** [ID] **& Christopher A. Schuh** [ID] [✉]

A comprehensive understanding of material strength across strain rates typically requires the combination of results from different methods, which often vary in loading conditions and/or sampled volumes, leading to discrepancies in material behavior. This study presents a microindentation approach to measure hardness covering eleven orders of magnitude in strain rate, from quasi-static to phonon drag-dominated rates, on a single material surface under uniform testing conditions. By engineering the geometry of impactors used in laser induced particle impact testing, we extend the breadth of accessible strain rates, including multiple distinct rates exceeding $10^5 \, \mathrm{s}^{-1}$. This self-consistent approach provides clearer insights into high-rate deformation mechanisms. Our results demonstrate a gradual increase in hardness with strain rate from quasi-static up to ultra-high rates, where a sharp upturn in hardness is observed.

In over a century of quantitative mechanical testing of materials, one overarching issue that the community has not traversed is that a full picture of materials strength requires an array of methodologies that address different conditions, which then need to be stitched together to form a complete picture. Such "stitching" has to span gaps in the measured quantity (such as comparing yield and flow strengths[1–4]), gaps in the applied stress states (such as comparing hardness and uniaxial strength[5]), gaps in the sampled volume (as between micromechanical tests and conventional ones[6]) and very large gaps in accessible strain rates (such as between quasi-static testing and high-rate impact-based experiments[7–9]). Combining such datasets also typically involves the compilation of measurements from different labs and operators, and different batches of material[3,6,10–13], giving large and uncontrolled uncertainties in the assembled picture.

One promising route to address this challenge lies in the evaluation of hardness as a unifying strength metric by combining indentation-based methods. Standard and advanced instrumented indentation techniques cover a broad portion of the strain rate spectrum[14,15], typically in regimes where thermally activated dislocation motion dominates[16]. Beyond this, laser-induced particle impact testing (LIPIT) has emerged over the past decade as a powerful technique for dynamic indentation at ultra-high strain rates ($\sim 10^8 \, \mathrm{s}^{-1}$)—well

above the reach of conventional micromechanical systems, where viscous drag dominates dislocation motion[17,18]. In LIPIT, microscale particles are accelerated to very high velocities—reaching several hundred meters per second—and impact onto material surfaces, producing localized plastic deformation[19–21]. Originally developed to study bonding phenomena and interfacial mechanics[22–24], LIPIT has more recently been applied to assess mechanical strength via the indentation-like craters left by impacting ceramic particles[7,8,25]. Nevertheless, a substantial strain-rate gap remains between instrumented indentation and conventional LIPIT, limiting the ability to construct a continuous and self-consistent picture of strength evolution. Bridging this gap requires extending LIPIT toward lower strain rates, which is non-trivial: reducing particle velocity also reduces impact energy and indentation depth.

Our purpose in this paper is to present the first study to establish a self-consistent set of mechanical strength measurements spanning a very large range of strain rates from the quasi-static range to the extreme rates at which phonon drag dominates deformation. We introduce new classes of LIPIT impactors that enable controlled deformation at lower velocities, allowing a more straightforward comparison between conventional LIPIT results with standard and impact indentation datasets. The self-consistency in our approach

Department of Materials Science and Engineering, Northwestern University, Evanston, IL, USA. [✉]e-mail: schuh@northwestern.edu

spans the dimensions described above: we provide measurements of indentation hardness with a single standard definition, at the same essential size scale (microhardness), on the self-same material surface, measured by a single operator.

## Results

### Defining hardness and strain rate

Our approach uses micro-indentation because it combines many advantages. First, it produces generalizable bulk strength (hardness) measurements with strong statistical support, and can be conducted in a high-throughput mode. Second, it requires only a small volume of material, and the same piece of material can be used for many conditions. Finally, with some innovation—as we shall see shortly—we are able to use this single test type over eleven orders of magnitude in strain rate. Hardness can be defined as the ratio of the plastic work expended ($W_p$) over the indentation volume ($V_{ind}$):

$$H = \frac{W_p}{V_{ind}} \qquad (1a)$$

For a dynamic indentation experiment, the plastic work is related to the kinetic energy dissipated by the impact of the indenter:

$$H = \frac{\frac{1}{2} m_i (v_i{}^2 - v_r{}^2)}{V_{ind}} \qquad (1b)$$

where $m_i$ represents the mass of the indenter/impactor and $v_i$ and $v_r$ are the impact and rebound velocities, respectively. Another equivalent version of Eq. 1a is more common when the applied load-displacement (P-h) curve of the indentation is known:

$$H = \frac{\int P\,dh}{V_{ind}} \qquad (1c)$$

The volume of the indentation crater can be recast as the integral of the cross-sectional area of the indenter over its depth, i.e., $V_{ind} = \int A_{ind}\,dh$[26]. Provided that there are no size effects on hardness, i.e., $\frac{dH}{dh} = 0$ (which holds for any material under a self-similar indenter geometry when the indentations are large enough to avoid material size effects, or for non-strain-hardening materials under any indenter), then Eq. 1c rearranges to:

$$H = \frac{P}{A_{ind}} \qquad (1d)$$

which is the classical Meyer definition of hardness; this version of Equation 1 has the advantage that only the maximum load of the indentation is needed, and can be set against a separate measurement of the projected contact area. Importantly, all four of the above expressions are equivalent rearrangements of one another, thus forming a self-consistent basis for the experiments to follow.

Beginning at quasi-static rates, we use standard methods of instrumented indentation across a range of applied strain rates. Of course, in both conventional indentation and impact experiments, the strain rate is a function of position and decreases as the indenter penetrates deeper into the material. Dimensional analysis requires that the average, or effective strain rate ($\dot{\varepsilon}$) is controlled by the ratio of impact velocity ($v_i$) to final indentation depth ($h$)[27]:

$$\dot{\varepsilon} = \frac{v_i}{h} \qquad (2)$$

A number of studies have evaluated a representative or average strain rate corresponding to the hardness values obtained from such experiments[27–33]. All of these conform to the scaling of Eq. (2), and are

thus equivalent to within a small prefactor that is generally not significant compared with the dominant ratio in Eq. (2). As a result, Eq. (2) provides a standard definition in line with widely accepted practices in nanoindentation, and we adopt it here.

### Standard and impact indentation

A conventional micro- or nanoindenter typically reaches strain rates up to $\sim 1\,\text{s}^{-1}$, and we begin by populating data from $10^{-3}\,\text{s}^{-1}$ up to that range for two single-phase materials, pure Ni and a solid solution of Ni-20 wt.%Cu. Nickel is chosen due to its widespread use and engineering relevance, making it an ideal candidate for exploring material behavior across a broad range of strain rates and providing validation touchpoints against prior literature. Ni-20Cu is selected as a complementary material to introduce compositional and mechanical differences while ensuring comparable fabrication methods. Single-indent data, represented by empty diamonds in Fig. 1, are collected using Eq. 1d by measuring applied loads and projected contact areas directly, while solid diamonds indicate average values from each set. Details of indentation protocols are outlined in Methods. Notably, constant loading rates were chosen, which means that the ratio of indentation velocity to depth ($\dot{h}/h$ or $v_i/h$) or, equivalently, loading rate to load ($\dot{P}/P$) decreases as indentation progresses, mimicking impact behavior (to which we will compare shortly). The data in Fig. 1 in this quasi-static range show a positive strain-rate dependence, which is a typical signature of thermally activated plastic flow[34], and the solid solution alloy is harder than the pure metal, as expected.

When the strain rate rises above ~$10-10^2\,\text{s}^{-1}$, inertial effects complicate instrumented indentation, making impact indentation methods more attractive. We advance to rates on the order of $10^3\,\text{s}^{-1}$ by using a common pendulum-based impact indentation method, as shown by the squares in Fig. 1, computed on the basis of measured velocities via Eq. 1b. These experiments were conducted using literally the same indentation tips as the previous quasi-static experiments

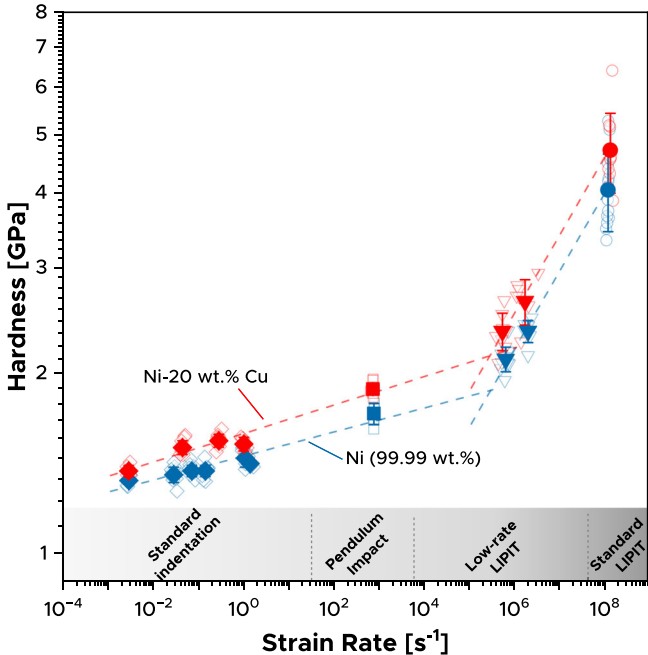

**Fig. 1 | Hardness as a function of strain rate $\dot{\varepsilon} = \frac{v}{h}$.** Data for pure 99.99% nickel (blue) and nickel with 20 wt.% Cu in solid solution (red). Empty symbols represent single experimental results, while solid ones indicate average values, and error bars represent the standard deviation of the hardness data. Diamonds refer to data acquired with standard instrumented indentation. Squares refer to data acquired with pendulum impact indentation. Triangles and circles indicate data generated using LIPIT by launching diamonds (modified low-rate LIPIT) and alumina particles (standard LIPIT), respectively. Source data are provided as a Source Data file.

(and the same substrates). Incorporating these results into Fig. 1 shows good alignment with quasi-static data for both materials. These findings extend the observed positive but modest strain-rate dependence of hardness, ascribed to thermally activated dislocation motion, to higher deformation rates.

### Accessing extreme strain rates with LIPIT

Beyond the rate of these data at $10^3$ s$^{-1}$, newer methodologies for dynamic indentation are needed. Significant efforts have been directed towards high-strain rate regimes, with some efforts achieving rates up to $10^4$–$10^5$ s$^{-1}$ [14,15,35,36]. Yet, accessing the ultrahigh strain rate domain (>$10^5$ s$^{-1}$) poses considerable challenges in both mechanical and electronic responses in classical instrumented indentation methodology. It is at this point that the major gap in Fig. 1 develops, because the next available technique to attain higher rates uses all-optical methods to bypass those mechanical and electronic responses [19,37,38]. LIPIT enables dynamic hardness measurements at extremely high strain rates [7] by launching a hard indenter/impactor towards a target material, typically at velocities of several hundred meters per second. A high-speed camera records the particle trajectory and impact, enabling the measurement of its velocity before and after impact. Following impact, the indentation volume can be determined by measuring the depression at the surface of the material, for example via laser confocal microscopy or atomic force microscopy and the hardness is then derived from Eq. 1b. In the present work, we have used this method to generate data at strain rates of ~$10^8$ s$^{-1}$ by launching alumina particles at high impact velocities (~$100$–$250$ m s$^{-1}$), as indicated by circles in Fig. 1. These data align with prior measurements by LIPIT [7,8,25,39], although we note here that we report the average strain rate according to the accepted definition of Eq. 2 with the indentation depth as a characteristic length scale, whereas prior works have used a different definition with the diameter or width of the impactor itself for that length scale. The present definition is more self-consistent with other indentation experiments and returns somewhat higher strain rates.

### Bridging the gap: low-rate LIPIT

Although LIPIT has gained popularity in micromechanics [8,25,40], it has been strictly limited to the most extreme strain rates: our value of $10^8$ s$^{-1}$ is typical, and it is not a simple matter to lower that strain rate. This is because the obvious pathway to lower the strain rate in impact experiments is to lower the impact velocity, but such a decrease drastically lowers the input energy (which goes as velocity squared, meaning that a tenfold decrease in velocity lowers the input energy by two orders of magnitude). This in turn leads to shallower indentation depths and lowers the sampled volume: this causes problems because the response can drift into the range of indentation "size effects" when the impacts are in the submicron range, and in the limit, reduces the amount of plasticity to the point where the impact is purely elastic. As a result, the average strain rate ($v_i/h$) remains nearly constant because the decrease in particle velocity ($v_i$) is offset by the shallower indentation depth ($h$). In other words, standard LIPIT currently operates exclusively in strain rate regimes about five orders of magnitude higher than those achieved by more conventional indentation techniques. To our knowledge, the combined standard instrumented indentation, pendulum impact, and standard LIPIT data in Fig. 1 represent the first time that those methods have been compared on the same test material, and the five-decade gap between them leaves major questions about the nature of the uptick in strain rate hardening seen there: a very broad interpolation is needed to stitch those data together.

Our proposed approach to bridge this large strain rate gap and cover strain rates ranging from $10^5$ s$^{-1}$ to $10^8$ s$^{-1}$ is to engineer the

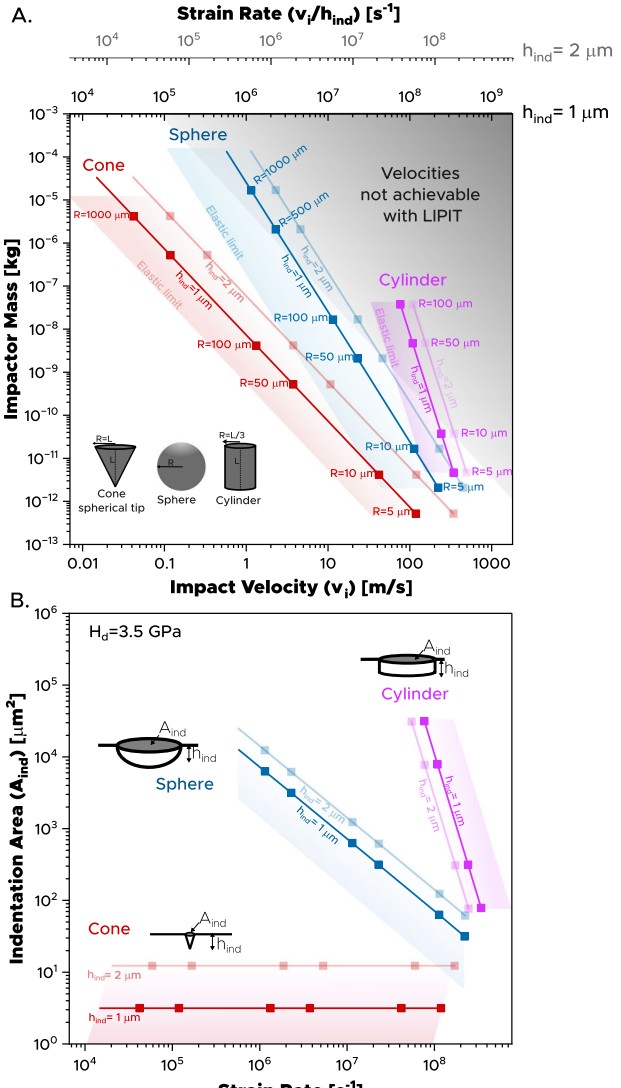

**Fig. 2 | Effect of impactor shape and mass on accessing lower strain rates in LIPIT. A** Mass versus incident velocity ($v_i$) of impactors impacting a material of hardness 3.5 GPa. Contours representing final indentation depth ($h_{ind}$) of 1 μm and 2 μm are illustrated for alumina impactors of conical, spherical and cylindrical shape. The faded gray region indicates the empirical power law that establishes an upper velocity limit based on impactor mass. **B** Indentation area ($A_{ind}$) as a function of strain rate for impacts considered in (**A**). Source data are provided as a Source Data file.

impactors used in LIPIT testing; by properly engineering the shape and mass of diamond impactors, we will establish a means of achieving lower strain rates while maintaining similar indentation depths. The challenge is laid out in Fig. 2A, which charts the relationship between mass and indentation depth for impactors of given velocities impacting on a target with a fixed hardness of $H = 3.5$ GPa; the corresponding strain rate is displayed on the upper $X$ axis. This fixed hardness value represents the dynamic hardness of the target material and is used solely as a reference to simplify the analysis, although it might change somewhat with strain rate for many test materials. This figure includes alumina indenters of various shapes, and considers an empirical power law that establishes an upper limit of impact velocity as a function of impactor mass reported in ref. 41. This empirical power law represents the situation in which an increase in laser energy no longer results in a corresponding increase in velocity due to the activation of other mechanisms of energy dissipation, such as launch pad fracture. When

using spherical particles, an order of magnitude decrease in strain rate requires an order of magnitude increase in particle size to achieve the same indentation depth (Fig. 2A-blue region); by extension, the indentation area expands considerably (as shown in Fig. 2B-blue region), moving the method beyond the micrometer scale. This illustrates the tradeoff inherent in reducing strain rates when using spherical particles: decreasing velocity and strain rate requires a very large increase in mass, which in turn increases the contact area and changes the scale of the test. This relationship of indenter mass, velocity and indentation depth becomes even worse when considering cylinders of aspect ratio three (Fig. 2-pink region): to lower the strain rate requires larger impactors at a steep slope. For spheres and cylinders, LIPIT testing is only plausible at rates over about $10^7$ or $10^8\,\mathrm{s^{-1}}$, respectively.

The key to extending LIPIT to lower strain rates is thus to break the poor scaling of mass and indentation area of spherical and cylindrical impactors, seen in Fig. 2B. Ideally, indenters with a small contact area and a large trailing mass should break this tradeoff. The simplest example is a self-similar sharp indenter geometry, for example, a cone, for which velocities (and strain rates) can be reduced while maintaining similar indentation areas (Fig. 2-red region). Figure 2A shows that sharp indenters can be scaled to velocities below $1\,\mathrm{m\,s^{-1}}$, attaining strain rates as low as $10^4\,\mathrm{s^{-1}}$, while Fig. 2B shows that this can be done at constant contact area and depth.

We therefore proceed to adapt LIPIT to use two such indenter geometries at various scales, including sharp polished diamond pyramids, along with in-house designed and microfabricated diamond impactors. Specific details of the LIPIT setup and impactors are provided in Methods. Our polished diamonds (Fig. 3A−C), have a self-similar sharp geometry (blunted to ~10 μm). We selectively removed portions of the back sides of these diamonds, to achieve a broad range of impactor masses varying from 90 to 700 μg. In parallel, microfabricated diamond impactors were fabricated from CVD diamond plates following a procedure described in ref. 42. A schematic overview of the process is outlined in Fig. 3D, along with scanning electron microscopy images of the final impactors. These microfabricated impactors feature a small circular flat punch as the indentation tip, and have a large trailing mass attached as a base, which contributes to the impactor mass without affecting the indentation area.

Figure 4A presents a series of snapshots showing a polished diamond, launched by LIPIT, approaching and impacting the nickel target at approximately $2.4\,\mathrm{m\,s^{-1}}$. Despite this relatively low velocity, signs of plastic deformation are already visible upon impact, as shown in frames 4 and 5. In Fig. 4B, a confocal image captures the resulting indentation crater and its profile, revealing a penetration depth of about 5 μm. Notably, the average strain rate during this impact event is $\sim5\times10^5\,\mathrm{s^{-1}}$, about three decades slower than the strain rate produced by conventional LIPIT. Similarly, snapshots in Fig. 4C show the impact event of a microfabricated diamond, containing a flat punch of $D\sim35\,\mu m$, impacting a Ni-20 wt.% Cu target at a velocity of $16\,\mathrm{m\,s^{-1}}$. The impact resulted in an indentation of $\sim5\,\mu m$ depth (Fig. 4D), representing an average strain rate of $3\times10^6\,\mathrm{s^{-1}}$. The laser-diced polished diamonds, displayed in Fig. 3A left, were launched at velocities comparable to those of the microfabricated diamonds, which permits us to directly establish that the details of the indenter shape are not critical to the measurement of hardness according to Eq. (1). The results align in terms of both the strain rate and resulting hardness (see Supplementary Information). The data points at strain rate of $\sim10^6\,\mathrm{s^{-1}}$ in Fig. 1 thus include tests conducted with both microfabricated and laser-diced polished diamonds. Similarly, additional experiments have been conducted to validate the strain rate definition for spherical indenters used in conventional LIPIT tests, as detailed in the Supplementary Information.

In general, speeds achieved with these diamonds were on the order of $1-20\,\mathrm{m\,s^{-1}}$, depending on the impactor mass, with indentation

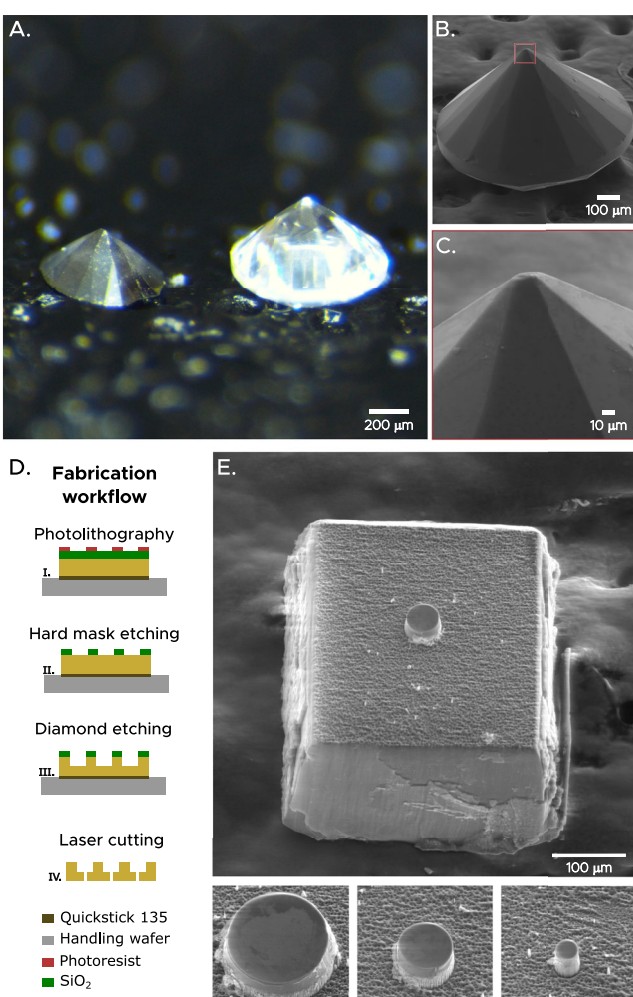

**Fig. 3 | Projectiles used in LIPIT to access lower strain rates. A** Image of a synthetic polished diamond pyramid with base 0.8 mm (right), alongside one with its height laser-diced to tailor mass (left). **B**, **C** SEM images of a synthetic polished diamond. **D** Fabrication workflow of microfabricated diamond impactors, following ref. 42. **E** SEM images of a microfabricated diamond impactor with a ~35 μm diameter flat punch, together with flat punches of varying size.

depth ranging from 3 to 15 μm. This demonstrates that by engineering LIPIT impactors, it is possible to access lower strain rates, while also showing that impactors that are 3–4 orders of magnitude heavier than conventional LIPIT microprojectiles can be successfully used in LIPIT. In this regard, it is worth mentioning that although LIPIT launch pad manufacturing has evolved to reach higher peak velocities, the power law function relating velocity to mass proposed in ref. 41 also applies at low velocities and heavy impactors (see Supplementary Information). This relationship serves as a valuable tool for designing LIPIT experiments in advance and thus for measuring hardness across the entire extreme strain rate regime.

These adapted low-rate LIPIT experiments produce the data shown with triangles in Fig. 1. These results nicely fill the large gap between the extreme conditions of conventional LIPIT and classical pendulum impact indentation. What is more, in the present study this bridging is done with the same indenter tips: the diamonds shown in Fig. 4A were mounted for all of the experiments below $10^6\,\mathrm{s^{-1}}$ in Fig. 1. All of the experiments in Fig. 1 also reflect measurements at the same size scale, with depths all in the range 2–15 μm; this is well outside of the range where size effects are normally seen[43], and thus all the present data represent bulk measurements.

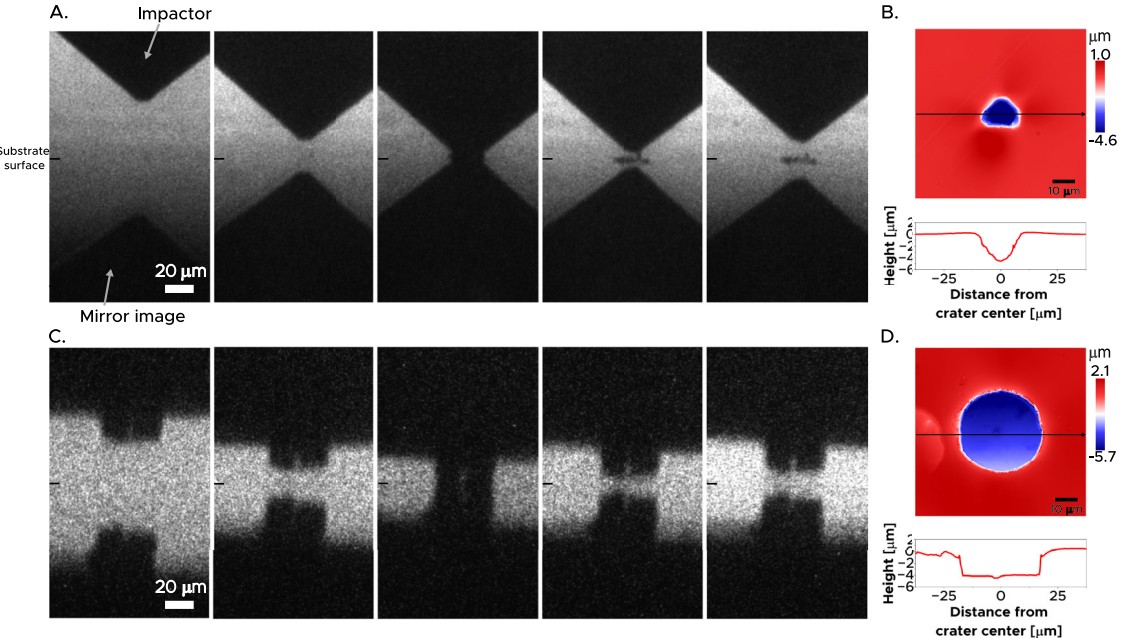

**Fig. 4 | In-situ observations of impact events using diamond impactors along with the corresponding confocal images. A** Snapshots of a polished diamond impacting a nickel target at 2.4 m s$^{-1}$. **B** The corresponding confocal image of the indent produced (**A**). **C** Snapshots of a microfabricated diamond impacting a Ni-20 wt.% Cu target at 22 m s$^{-1}$. **D** The corresponding confocal image of the indent produced in (**C**).

## Discussion

The combination of LIPIT experiments with both instrumented quasi-static and impact indentations allows the evaluation of hardness across the entire range of strain rates, from quasi-static ($10^{-3}$ s$^{-1}$) to ultra-dynamic ($10^{8}$ s$^{-1}$). The data are self-consistent on all the axes that have created difficulty in stitching test data together in the past: they use a single definition of hardness and strain rate, they are all at the same physical size scale, they are all from the same material surface, collected by a single operator. Such data enables quantitative discussions about the mechanisms and constitutive models for plasticity with higher certainty. For example, the data in Fig. 1 reflect two very distinctive regimes of strain rate sensitivity ($m = \frac{\partial \ln(H_d)}{\partial \ln(\dot{\varepsilon})}$). At strain rates below $10^{4}$ s$^{-1}$ both Ni and Ni-20Cu exhibit low values of $m_{Ni} = 0.021$ and $m_{Ni-20Cu} = 0.024$, in line with those reported for Ni in the literature[34,44]. In contrast, at strain rates exceeding $10^{5}$ s$^{-1}$, both materials demonstrate significantly higher strain rate sensitivities of $m \sim 0.13$. This transition has been seen before in refs. 3,7,25, but required stitching data together from independent studies with different measurements of strength. Similarly, experiments over narrow strain rate ranges in refs. 2,5,45 have suggested such an upturn in rate hardening as well, but it has been considered disputable on the basis of data limitations[12]. The present fully self-consistent dataset strongly supports a narrow transition range between two hardening rates. As discussed in prior works, plasticity can produce higher dislocation contents at these higher rates[46–49], although changes in the thermal activation character of flow in the high-rate regime[40,50] suggest a transition in the predominant deformation mechanism shifting from thermally activated dislocation motion to dislocation drag, or viscous drag. With more data over several decades of strain rate, our measured strain rate sensitivity at these high rates is more certainly measured at $\sim 0.13$, as compared to that inferred from sparse prior data at ~0.2 in earlier studies[45].

The dominant mechanism in this high-strain-rate range, viscous drag, can be understood as a dissipative force proportional to the dislocation velocity, and therefore to the strain rate; the dominant expected dissipative force is dislocation-phonon interactions. In this regime, the strength can be described as scaling with a macroscopic viscosity ($\eta$) and strain rate, expressed as $\sigma \propto \eta \dot{\varepsilon}$[17]. With our data in Fig. 1, we are able to directly estimate the macroscopic viscosity for nickel as ~18 Pa s from the slope of a linear hardness–strain rate plot at strain rates exceeding $10^{5}$ s$^{-1}$. This viscosity can be written as $\eta = B/\rho_m b^2$[17,18], where $B$ is the drag coefficient or damping constant, $\rho_m$ is the density of mobile dislocations, and $b$ is the Burgers vector. Given a mobile dislocation density of $\sim 10^{13}$ m$^{-2}$ under these high deformation rates according to ref. 51, and a Burgers vector of 2.5 Å, the drag coefficient is extracted from our data as 1.1 10$^{-5}$ Pa s. This value aligns well with theoretical estimates of phonon drag coefficient (which range from $\sim 10^{-6}$–$10^{-5}$ Pa s[52]) and is comparable to the drag coefficient obtained from molecular dynamics simulations of edge dislocations ($\sim 10^{-5}$–1.5 10$^{-5}$ Pa s[53,54]). Such experimental assessment of the drag coefficient was not supported by classical LIPIT data alone, and its investigation using other techniques is challenging, as the data typically span a narrow strain rate range with significant dispersion. The present experimental findings thus support the notion that the dominant deformation mechanism at these extreme rates is dislocation-phonon damping, and offer a view to quantitatively elaborating it. We note that many existing models tend to align with experimental results in either the thermal activation or the drag-dominated regimes[55–57]. This is at least partly due to the lack of self-consistent experimental data spanning the entire strain rate range. By addressing this gap, the techniques in this paper can provide the foundation for more comprehensive and accurate constitutive modeling in the future.

In summary, the current study represents, to our knowledge, the first use of self-consistent micro-indentation-based techniques to measure material strength or hardness across such a wide range of strain rates, including multiple distinct rates exceeding $10^{5}$ s$^{-1}$. This was enabled by engineering the geometry of LIPIT impactors to widen the range of accessible strain rates, all while preserving microscale characteristics. By coupling this approach with a self-consistent suite of conventional instrument indentation methods and conventional LIPIT, 11 decades of strain rate can now be effectively explored without large gaps, on the same material surface, with the same definitions and measures of hardness across the full spectrum. By providing new

quantitative clarity on extreme deformation mechanisms like phonon drag plasticity, we hope the technique paves the way for significant micromechanical investigations in extreme conditions, while also smoothly connecting to classical behaviors at more conventional test rates.

## Methods

### Sample fabrication

Nickel (99.995%) and copper pellets (99.999%) (MSE Supplies, US) were employed to prepare pure nickel and Ni-20 wt.% Cu specimens through arc melting (compact model MAM-1; Edmund Buhler, Germany) in an argon atmosphere. The resulting buttons were held at 600 °C for two hours and then cooled to room temperature in the furnace. Following this, samples were cut and conventionally ground and polished to achieve a 0.01 μm surface finish.

### Instrumented indentation experiments

Quasi-static and impact indentation tests were conducted using the NanoTest Alpha instrumented nanoindenter from Micro Materials Ltd. (UK). Details of this pendulum-based instrument can be found elsewhere[58]. A polished pyramidal diamond tip, the same as used in LIPIT (see details below), was used for all indentations, with protocols selected to achieve indentation depths exceeding three microns.

For the quasi-static indentations, constant loading rates of 15, 25, 50, 100, and 400 mN s$^{-1}$ were applied up to a peak load of 490 mN. This strategy of constant loading rates was chosen since the ratios of velocity to depth or loading rate to load, mimic the behavior during impact indentations, meaning that $v/h$ decreases as the indentation progresses. The indentation velocity was computed from the slope of the displacement-time plot. The indentation depth and indentation area were measured using 3D laser scanning confocal microscopy (VK-X200 Keyence, Japan). The average strain rate was determined as the ratio of velocity to indentation depth. The hardness under quasi-static conditions was calculated using Eq. 1d.

In the case of pendulum impact tests, an impulse distance of 25 μm and impulse force of 30 mN were employed. Equation 1b was considered for the impact data. The pendulum effective mass was calibrated by conducting a series of experiments at fixed retraction distance (10 μm) and variable impact forces (1–10 mN), following the procedure introduced in ref. 59. The resulting displacement-time data were differentiated twice to obtain acceleration profiles before impact, and the effective mass was determined from the slope of the impact force versus acceleration plot. Similarly, the incident and rebound velocities were extracted by fitting and differentiating the displacement-time data, as outlined in ref. 31. Laser scanning confocal microscopy was employed to determine the post-mortem impact depth and volume, which were used to calculate the strain rate and hardness, respectively.

### LIPIT experiments

Micromechanical impact experiments at high deformation rates were conducted using an in-house-designed microscale ballistic platform to accelerate individual impactors; a general description of the setup can be found in ref. 60. Briefly, an intense laser pulse was focused onto a launch pad containing impactors; upon ablation of a metal film, impactors were accelerated toward the target material. Stiff, glass-on-glass launch pads introduced in ref. 60 were employed. This helped to maintain a minimal launch pad-to-target gap (around 100 μm), which not only reduced the effect of impactor misalignment but also helped fully capture the indentation event on camera, even at slower velocities compared to conventional LIPIT. Prior to the laser pulse, and when they were sitting on the launch pad, diamond impactors were positioned in all cases so that only a small portion of the tip/punch appeared at the top of the imaging frame. The impactor velocity was varied by modifying the laser energy, and a high-speed camera (SIMX16, Specialized Imaging, US) was employed to track the impactor trajectory and analyze the impact and rebound velocities. In this study, velocities ranged from 1 m s$^{-1}$ to 250 m s$^{-1}$ to stay below the jetting/spall regime. The impact depth and volume were examined by laser scanning confocal microscopy.

### LIPIT impactors

Synthetic chemical vapor deposition (CVD) diamonds, featuring a base diameter of 0.8 mm and polished to a pyramidal shape with approximately 56 facets, were acquired from Starsgem Co. (China). The final height was, in some cases, modified by laser-guided waterjet (Evolved Diamonds LLC, US) to tailor the impactor mass (Fig. 2A–C). These sharp indenters have an area function matching a sharp cone with an angle of 115°.

For the case of microfabricated diamond impactors (Fig. 2E, D), the overall process flow is outlined in ref. 42. In brief, the process began by cleaning a monocrystalline CVD diamond plate (7 × 7 × 0.3 mm$^3$) polished to a 5–10 nm Ra finish (Evolve Diamonds LLC, US) using Piranha solution. The plate was subsequently cleaned in an oxygen plasma, and a 2 μm-thick silicon oxide hard-mask layer was deposited through plasma-enhanced CVD immediately afterwards. A positive photoresist was spin-coated, and a predefined pattern was transferred by photolithography. A dry etching of the silicon dioxide hard mask was conducted to transfer the pattern to the hard mask and was followed by a diamond etching in an inductively coupled oxygen plasma until the hard mask was completely consumed. Finally, squares with centered flat punches were diced around each flat punch using laser dicing. Further details regarding the etching conditions can be found in the Supplementary Information.

Spherical alumina microparticles (Inframat Advanced Materials LLC, USA), 20 ± 1 μm nominal diameter, were selected to create plastic indents at high impact velocities (∼100–250 m s$^{-1}$). No evidence of particle deformation or fracture was observed upon impacts, for which all plasticity is assumed to be confined to the substrate.

For the case of diamonds, the mass of each impactor was measured using a piezoelectric crystal microbalance with a resolution of 1 μg, while for alumina particles, the mass of each of them was calculated based on their measured diameters prior to experiment. In both cases, it was assumed that all plastic deformation was confined to the substrate.

### Reporting summary

Further information on research design is available in the Nature Portfolio Reporting Summary linked to this article.

## Data availability

The authors declare that the data supporting the findings of this study are available within the paper, their supplementary information and source data files. Source data are provided with this paper.

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

## Acknowledgements

The authors thank Steven Kooi at MIT for his design, assembly, and ongoing support of the current LIPIT setup. This work was primarily supported by the US Department of Energy, Office of Science, Office of Basic Energy Sciences, Division of Materials Sciences and Engineering under awards DE-SC0018091 and DE-SC0025282. LB acknowledges the support of the SNSF through Postdoc.Mobility fellowships P500PN_217723. This work made use of the NUFAB facility of Northwestern University's NUANCE Center, which has received support from the SHyNE Resource (NSF ECCS-2025633), the IIN, and Northwestern's MRSEC program (NSF DMR-2308691).

## Author contributions

L.B. and C.A.S. proposed the research idea, discussed the results, and contributed to writing and reviewing the manuscript. L.B. performed all the experiments. C.A.S. provided overall guidance.

## Competing interests

The authors declare no competing interests.
