## [Transparent Peer Review file · Nature Communications]

Self-consistent hardness measurements spanning eleven decades of strain rate on a single material surface

Corresponding Author: Professor Christopher Schuh

Version 0:

Reviewer comments:

Reviewer #1

(Remarks to the Author)

This study has challenged the development of LIPIT having unique projection of conical and various indenters etc. Experimental set up is very unique, the topic is new (wide range of strain rate on hardness) and new finding of phonon drag of dislocation motion is discussed. Therefore, it will be published in this journal before the following questions be addressed.

Around equations (1) and (2) are very rough, and the mechanics of strain rate is skipped. Indentation generally induces inhomogeneous strain and strain rate. Thus, we need to take a representative strain and strain rate. For example, FEM is useful (but constitutive equation is needed). The following papers can be cited. Could you check your approach for verification to calculate indentation strain rate ?
<https://doi.org/10.1016/j.commatsci.2015.08.033>
<https://doi.org/10.1007/s11665-022-07507-8>

When the projectile velocity is slow, the mass of indenter should be large in order to realize deep penetration. Which is the dominant factor of plastic deformation of LIPIT impact, indenter velocity or indenter mass? FEM can simulate this issue (as continuous mechanical model), but experiment is difficult since special impactor (like this study) is needed. Thus, this point can be emphasized in this study.

Experiment of single diamond indenter flying is very challenging. Fig 4 shows snapshot of impact indenter image at free surface. Are there any image for straight flying of indenter. The flying distance may be from 1 to 2 mm.

Why don't you predict the line of hardness-strain rate ? via active energy and phonon drag of dislocation motion ? FEM and theory can be applied if you know constitutive equation.

Around line 230, there is typo or mis conversion.

Reviewer #2

(Remarks to the Author)

This study aims to develop self-consistent hardness measurements from low to high strain rates using combined micro-indentation and LIPIT experiment. By self-consistency authors mean probing the same amount of material volume from the same surface, performed by the same operator with the same definition of hardness. Finally, these measurements are discussed in the context of standard dislocation drag and thermally activated dislocation motion models.

1) The primary concern about this study is that the strain rate metric should accurately reflect the gradient of velocity within the material during deformation under impact. Since the impactors used in the experiments differ not only in material but also significantly in geometry, the influence of these geometrical differences on the velocity gradient of the material remains unclear. As a result, defining strain rate in the same manner for different indenter geometries in LIPIT experiments raises questions about the validity of this approach and the claim of self-consistency. Given that authors use LIPIT experiments to probe dislocation drag mechanisms, ensuring consistency in strain rate across experiments with different geometries is

critical. Additionally, variations in the small elastic deformations of the impactors, such as the sharper conical diamond indenter, the cylindrical diamond indenter, and the alumina sphere could further influence the indentation strain rates. To address these issues, the authors should demonstrate that the strain rate definition remains consistent irrespective of different geometries and materials by either simulations incorporating material models for the indenters to compare strain rate fields across the geometries or by LIPIT experiments comparing hardness values across different indenter geometries at approximately the same strain rate range in the dislocation drag regime.

2) In Figure 2, the maximum velocity for a given mass appears to be dependent on the specific LIPIT system. What affects it?

3) Lines 219-221: Is the phonon drag the sole explanation for the observed transition? There seems to be no direct observations of dislocation-phonon interactions and it appears that this result is derived primarily from plasticity models. There are studies that observed dislocation multiplication consistent with the hardness trends, suggesting a potential role for dislocation nucleation. The authors are encouraged to expand on alternative mechanisms that could contribute to the sharp increase in rate sensitivity.

Reviewer #3

(Remarks to the Author)

Please see attached pdf for comments

Version 1:

Reviewer comments:

Reviewer #1

(Remarks to the Author)

The authors have answered all comments well. Congratulations.

Reviewer #2

(Remarks to the Author)

While previous reviewer comments have been addressed to some extent, the concern about self-consistency of strain rate definition among different impactors remains inadequately resolved, that authors should resolve before publication:

The reviewers agree that the definition of hardness is self-consistent for conical and cylindrical diamond indenters. However, no such consistency has been demonstrated between the microfabricated impactors used for strain rates of 10^5 - 10^6 s⁻¹ and the alumina spheres used for strain rates of 10^8 s⁻¹. Since the model beyond the rate-sensitivity upturn (pages 10–11) relies on experimental results from both diamond and alumina impactors, and is highly sensitive to strain rate, it is crucial to establish a consistent strain rate definition across all impactors, regardless of their geometry or material. Without this, any attempt to develop a model would lack a solid experimental foundation.

Reviewer #3

(Remarks to the Author)

The details provided in the reviewer's response have made the paper much stronger. I think this paper is worthy of publication, though I still feel that the strongest contribution is in the areas of LIPIT and showing the different geometric control, which does not perhaps merit publication in Nature Communications. Indeed, their claim that LIPIT has never been connected to mainstream testing methods (response to question 3, reviewer 3) is incorrect (for example, ACS Applied Materials Interfaces, 2023, 15, 32916-32935 for connection to experimental methods, ACS Macro Letters, 2024, 13, 3, 302-307 for connection to MD simulation). In addition, I still feel that the authors are relying on only a few data points to claim a universal curve over 11 orders of magnitude (Figure 2). While they addressed this point in their comments to the reviewers, I still believe, even with the experimental difficulty, they could fill in some of the areas, in particular in the 10^0 - 10^4 regime. I think to merit such bold claims, a few more data points in the intermediate regime are necessary. The authors have written a good paper, and I think it will have significance to the LIPIT and other high strain rate test community, but I do not think it rises to the level of universality they are claiming.

Version 2:

Reviewer comments:

Reviewer #2

(Remarks to the Author)

Authors have addressed this reviewer's comments reasonably well. Therefore, I recommend accepting this paper for publication.

Reviewer #1 (Remarks to the Author):

This study has challenged the development of LIPIT having unique projection of conical and various indenters etc. Experimental set up is very unique, the topic is new (wide range of strain rate on hardness) and new finding of phonon drag of dislocation motion is discussed. Therefore, it will be published in this journal before the following questions be addressed.

- 1) Around equations (1) and (2) are very rough, and the mechanics of strain rate is skipped. Indentation generally induces inhomogeneous strain and strain rate. Thus, we need to take a representative strain and strain rate. For example, FEM is useful (but constitutive equation is needed). The following papers can be cited. Could you check your approach for verification to calculate indentation strain rate ? <https://doi.org/10.1016/j.commatsci.2015.08.033>
<https://doi.org/10.1007/s11665-022-07507-8>

Authors' response: Thank you for your positive comments in the preamble.

We agree with the reviewer's insight regarding the inhomogeneous nature of the strain and strain rate in indentation and impact experiments. Indeed, in impact (or in indentation) experiments, the strain rate decreases as the indenter penetrates deeper into the material, and even then the strain rate at any instant is a function of position. Fortunately, this is not a new topic and there are decades of work establishing representative or average strain rates corresponding to the hardness values obtained from such experiments (including the references the reviewer has suggested). Even more fortunately, all of these efforts give the same scaling laws, meaning that although different choices of a strain rate definition are possible, they are all the same to within a small geometrical prefactor to the definition we have used. We have adopted the strain rate definition most widely accepted in the instrumented indentation community.

Changes to the manuscript: We have incorporated the references suggested by the reviewer and we have extended our explanation prior to and after Equation 2.

"Beginning at quasistatic rates, we use standard methods of instrumented indentation across a range of applied strain rates. Of course, in both conventional indentation and impact experiments, the strain rate is a function of position, and decreases as the indenter penetrates deeper into the material. Dimensional analysis requires that the average, or effective strain rate ($\dot{\epsilon}$) is controlled by the ratio of impact velocity (v_i) to final indentation depth (h)¹⁵:

$$\dot{\epsilon} = \frac{v_i}{h} \text{ (Equation 2)}$$

A number of studies have evaluated a representative or average strain rate corresponding to the hardness values obtained from such experiments¹⁵⁻²¹. All of these conform to the scaling of Eq. (2), and are thus equivalent to within a small prefactor that is generally not significant compared with the dominant ratio in Eq. (2). As a result, Eq. (2) provides a standard definition in line with widely accepted practices in nanoindentation, and we adopt it here."

- 2) When the projectile velocity is slow, the mass of indenter should be large in order to realize deep penetration. Which is the dominant factor of plastic deformation of LIPIT impact, indenter velocity or indenter mass? FEM can simulate this issue (as continuous mechanical model), but experiment is difficult since special impactor (like this study) is needed. Thus, this point can be emphasized in this study.

Authors' response: The initial work of indentation is proportional to $m v^2$. Thus, a tenfold decrease in velocity would require a hundredfold increase in mass to maintain the same input energy. This demonstrates that velocity plays a dominant role in driving plastic deformation. We agree that this interplay between velocity and mass is a critical point, we added a few sentences.

Changes to the manuscript: We have introduced a sentence in the first paragraph after Fig. 1.

"...This is because the obvious pathway to lower the strain rate in impact experiments is to lower the impact velocity, but such a decrease drastically lowers the input energy (which goes as velocity squared, meaning that a tenfold decrease in velocity lowers the input energy by two orders of magnitude)."

-
- 3) Experiment of single diamond indenter flying is very challenging. Fig 4 shows snapshot of impact indenter image at free surface. Are there any image for straight flying of indenter. The flying distance may be from 1 to 2 mm.
-

Authors' response: Indeed, experiments with large impactors present significant challenges and require modifications with respect to conventional LIPIT setups. For clarification, prior to the laser pulse and when they were sitting on the launch pad, diamond impactors were positioned in all cases so that only a small portion of the tip/punch appeared in the frame; however, due to the high height of impactors and the relatively small field of view of the camera, the base of impactors could not be captured in the image. For our experiments, we used glass-on-glass launch pads developed in <https://doi.org/10.1002/smt.202201028>, this helped us to keep a minimal launch pad-to-target gap, which was in all cases of a few micrometers (around 100 μm). This reduced any potential effect of misalignment. Additionally, the low gap allowed us to capture the full event within the camera's field of view; a very "long" exposure would otherwise be required due to the slower velocities involved.

Changes to the manuscript: We have extended our explanation of the experiments in the section "Methods-LIPIT experiments"

"...This helped to maintain a minimal launch pad-to-target gap (around 100 μm), which not only reduced the effect of impactor misalignment but also helped fully capture the indentation event on camera, even at slower velocities compared to conventional LIPIT. Prior to the laser pulse, and when they were sitting on the launch pad, diamond impactors were positioned in all cases so that only a small portion of the tip/punch appeared at the top of the imaging frame."

-
- 4) Why don't you predict the line of hardness-strain rate ? via active energy and phonon drag of dislocation motion ? FEM and theory can be applied if you know constitutive equation.
-

Authors' response: We have chosen to focus solely on the experimental results in this work, as we believe that the improvement of constitutive models across a wide range of strain rates will follow as a natural consequence of these findings. We note, however, that existing models tend to align with experimental results in either the thermal activation or the drag-dominated regimes, primarily due to the lack of self-consistent experimental data spanning the entire strain rate range. By addressing this gap, our study aims to provide the foundation for more comprehensive and accurate constitutive modeling in the future.

Changes to the manuscript: We have incorporated this into the discussion of the implications of our work.

"...The present experimental findings thus support the notion that the dominant deformation mechanism at these extreme rates is dislocation-phonon damping, and offer a view to quantitatively elaborating it. We note that many existing models tend to align with experimental results in either the thermal activation or the drag-dominated regimes⁴⁹⁻⁵¹. This is at least partly due to the lack of self-consistent experimental data spanning the entire strain rate range. By addressing this gap, the techniques in this paper can provide the foundation for more comprehensive and accurate constitutive modeling in the future."

-
- 5) Around line 230, there is typo or mis conversion.
-

Authors' response: Thank you for pointing this out, there seems to be a rendering problem.

Changes to the manuscript: We have corrected the line.

"...This viscosity can be written as $\eta = B/\rho_m b^2$ ^{43,44}, where B is the drag coefficient or damping constant, ρ_m is the density of mobile dislocations, and b is the Burgers vector."

Reviewer #2 (Remarks to the Author):

This study aims to develop self-consistent hardness measurements from low to high strain rates using combined micro-indentation and LIPIT experiment. By self-consistency authors mean probing the same amount of material volume from the same surface, performed by the same operator with the same definition of hardness. Finally, these measurements are discussed in the context of standard dislocation drag and thermally activated dislocation motion models.

1) The primary concern about this study is that the strain rate metric should accurately reflect the gradient of velocity within the material during deformation under impact. Since the impactors used in the experiments differ not only in material but also significantly in geometry, the influence of these geometrical differences on the velocity gradient of the material remains unclear. As a result, defining strain rate in the same manner for different indenter geometries in LIPIT experiments raises questions about the validity of this approach and the claim of self-consistency. Given that authors use LIPIT experiments to probe dislocation drag mechanisms, ensuring consistency in strain rate across experiments with different geometries is critical. Additionally, variations in the small elastic deformations of the impactors, such as the sharper conical diamond indenter, the cylindrical diamond indenter, and the alumina sphere could further influence the indentation strain rates.

To address these issues, the authors should demonstrate that the strain rate definition remains consistent irrespective of different geometries and materials by either simulations incorporating material models for the indenters to compare strain rate fields across the geometries or by LIPIT experiments comparing hardness values across different indenter geometries at approximately the same strain rate range in the dislocation drag regime.

Authors' response: We thank the reviewer for the feedback.

As pointed out in our response to reviewer 1 (point 1), strain and strain rate evolution in conventional indentation, and especially in impact experiments, are inherently inhomogeneous, allowing only average representations.

Regarding indenter geometries, it is critical to emphasize that we used the exact same lab-grown polished diamond indenters from quasistatic conditions up to strain rates of $\sim 10^6 \text{ s}^{-1}$. The inclusion of the microfabricated cylindrical indenters does not expand that range of strain rates, per se, because they are also employed at $\sim 10^6 \text{ s}^{-1}$ and are included in this study to show that: (i) diamond indenters can be microfabricated using conventional cleanroom techniques, and (ii) our testing technique is not limited to a single geometry tip, and in fact two different tip shapes give the same result. Data points generated using the microfabricated indenters align with those obtained using laser-diced polished diamonds in terms of impact velocity, computed strain rate, and resulting hardness (see below a zoom-in of data for nickel at $\sim 10^6 \text{ s}^{-1}$). This confirms, as the reviewer mentions, that strain rate and hardness values across different indenter geometries are equivalent within this regime. This data is now added to the Supplementary Materials.

Changes to the manuscript: We have extended our explanation in the first paragraph following Fig. 3 to clarify that laser-diced conical diamond and microfabricated diamond generated similar strain rate values and yielded comparable hardness results. We have also added a new figure to the supplementary materials.

"...The impact resulted in an indentation of ~ 5 μm depth (Figure 4D), representing an average strain rate of $3 \times 10^6 \text{ s}^{-1}$. The laser-diced polished diamonds, displayed in Figure 3A left, were launched at velocities comparable to those of the microfabricated diamonds, which permits us to directly establish that the details of the indenter shape are not critical to the measurement of hardness according to Eq. (1). The results align in terms of both the strain rate and resulting hardness (see Supplementary Materials). The data points at strain rate of $\sim 10^6 \text{ s}^{-1}$ in Figure 1 thus include tests conducted with both microfabricated and laser-diced polished diamonds."

2) In Figure 2, the maximum velocity for a given mass appears to be dependent on the specific LIPIT system. What affects it?

Authors' response: In LIPIT, to increase the impactor velocity, we typically increase the laser power. For a given impactor mass and launch pad configuration, the maximum velocity follows a power law relationship, as reported in <https://doi.org/10.1016/j.ijimpeng.2019.103465>. This relationship reflects the balance between the applied laser energy and the mechanical integrity of both the launch pad and impactor. Increasing the laser energy does not always result in higher particle velocities. Beyond a certain threshold, the high power starts to cause fracture of the glass of the launch (or even the impactor) and this limitation defines the maximum achievable velocity. Therefore, the launch pad configuration is the primary factor affecting the maximum achievable velocity.

Changes to the manuscript: We have added a sentence pointing this out in the second paragraph after Fig. 1.

"...This figure includes alumina indenters of various shapes, and considers an empirical power law that establishes an upper limit of impact velocity as a function of impactor mass reported in Ref. ³³. This empirical power law represents the situation in which an increase in laser energy no longer results in a corresponding increase in velocity due to the activation of other mechanisms of energy dissipation, such as launch pad fracture"

3) Lines 219-221: Is the phonon drag the sole explanation for the observed transition? There seems to be no direct observations of dislocation-phonon interactions and it appears that this result is derived primarily from plasticity models. There are studies that observed dislocation multiplication consistent with the hardness trends, suggesting a potential role for dislocation nucleation. The authors are encouraged to expand on alternative mechanisms that could contribute to the sharp increase in rate sensitivity.

Authors' response: Thank you for your comment. We recognize that dislocation multiplication could also contribute to the observed transition. We note, however, that even in this condition, the average dislocation velocity is limited by drag [https://doi.org/10.1016/0001-6160\(88\)90030-2](https://doi.org/10.1016/0001-6160(88)90030-2). We added mention of this alternative mechanism to the main manuscript.

Changes to the manuscript: Following the Reviewer's suggestion we have expanded on alternative mechanism that could contribute to the upturn phenomenon in our discussion of the results.

"...The present fully self-consistent dataset strongly supports a narrow transition range between two hardening rates. As discussed in prior works, plasticity can produce higher dislocation contents at these higher rates ³⁸⁻⁴¹, although changes in the thermal activation character of flow in the high-rate regime ^{32,42} suggest a transition in the predominant deformation mechanism shifting from thermally activated dislocation motion to dislocation drag, or viscous drag."

Reviewer #3:

General questions/comments:

1) -Overall, this is a nice paper with a very sound premise: try to use one unifying method to assess the materials properties across a large range of strain rates, with a particular interest in how to achieve moderately to ultra-high strain rates. It nicely blends different techniques into one equivalent mathematical formula to connect different results

- This manuscript claims to study hardness over 11 orders of magnitude. While technically correct, this statement is somewhat misleading. It characterizes the quasi-static regime, 10^{-3} - 10^0 , then has a single data point at 10^3 , then two data points at 10^6 , then another at 10^8 . This means that over 6 orders of magnitude, there is a single data point, which means interpolation across this regime is highly dependent on that single data point. Furthermore, the high strain rate regimes are also a series of three points. It is not that I do not believe these points or these trends, but I do not think there is enough data to fully support their claims across these regions.

Authors' response: We thank the reviewer for their time and feedback.

Regarding the concern about our claim to study hardness over 11 orders of magnitude, we respectfully disagree with the reviewer's assessment. As noted in the reviewer's first comment, the goal of our work is to present a unifying method for assessing material properties from quasistatic to ultra-high strain rate regimes. The number of data points in intermediate regions is inherently linked to the current experimental bandwidth of the laboratory, but critically, prior to this study there was literally no data in the intermediate regime around 10^6 s⁻¹ by ANY mechanical test method, and therefore a large gap that spanned the entire literature. The density of data we show here could, of course, be higher with more time and experimentation, and it is a tautology that more data is always better. Now that we have shown how it can be done, we fully expect that the field will react and produce more data. However, the key achievement in this paper is the broad bridging of these gaps in a single self-consistent study, which has never been done before and which makes this paper a significant moment for the field of mechanical testing.

(Parenthetically, interpolation across the remaining much smaller gaps in this data is quite sound, mathematically and mechanistically.)

2) All discussion is lifted from previous papers: validating and perhaps refining slightly what has been seen already with one test, but not adding anything new in terms of understanding what is causing the hardness

Authors' response: With respect, we disagree strenuously with this comment. There is no content in this paper that has been "lifted" from anywhere else: it is 100% original text and graphics, 100% new data, and developed in its own context. To the extent that there is commonality between this work and any others, it is because there is mechanistic consistency between this work and prior (more range-limited) works. Such consistency is desirable in science, and lends credibility to the current results. What the reviewer criticizes as "validating and refining" prior results we view as a critical requirement of a worthy contribution. Additionally, what the reviewer misses is that the present paper does more than "validate" prior work: it dramatically expands upon it and in fact provides undergirding over the broadest consistent set of mechanical data yet assembled across such a wide range of strain rates.

We believe the novelty of our work and approach is clearly demonstrated in the manuscript. The mechanical tests presented are significant results in their own, as they establish a unified framework for probing material behavior across an unprecedented range of strain rates.

3) In my opinion, the most novel part of the paper is a methodological development for LIPIT, namely, that using different shaped impactors can control the strain rate, is an important finding for the LIPIT community, but does not have obvious significance for others outside. In the world of puncture mechanics, such in studying needles, the shape of an indenter, albeit at lower strain rate than LIPIT, has been known to affect

the response of a material. So this is a useful finding, but not necessarily novel. Thus, I believe this paper merits publication, but not necessarily in Nature Communications, and not without significant revision

We certainly agree with the reviewer that the LIPIT community will find this work relevant, as our method substantially expands the capability of that technique. However, we respectfully believe that we have shown implications well beyond LIPIT by developing self-consistency of LIPIT with two other mainstream methods of nanomechanical testing in rigorous detail, in regimes that are far from what LIPIT can do. Historically, the study of materials across vastly different strain rate regimes has required different—and often difficult to compare—experimental techniques. Our paper makes the first true connection of LIPIT to mainstream testing methods, validating that entire line of research. The combined power of quantitative data across this full range of strain rates offers, for instance, a new direction for the calibration of strength models and understanding of mechanistic crossovers. Moreover, they establish a foundation for further studies, such as those investigating the microstructural aspects mentioned by the reviewer in the previous point, offering significant opportunities to advance materials design.

4) Some Specific questions/comments:

- Why were Ni and Ni-20Cu chosen for this study?
-

Authors' response: Nickel was chosen due to its widespread use and engineering relevance, making it an ideal candidate for exploring material behavior across a broad range of strain rates. There is also much published data on Ni (from many disparate techniques, labs and authors) to allow benchmarking with the literature. Since the technique is novel, it is important to validate it on a common material, and Ni is a good choice in that regard; note that we make several direct numerical connections to prior work on Ni in the paper. Ni-20Cu was selected as a complementary material to introduce slight compositional and mechanical differences while ensuring comparable fabrication methods. This allowed us to demonstrate that the proposed approach can effectively resolve the material properties of both materials consistently across the entire strain rate regime.

Changes to the manuscript: We have now included a sentence to cover these choices in the paper.

"...A conventional micro- or nanoindenter typically reaches strain rates up to $\sim 1 \text{ s}^{-1}$, and we begin by populating data from 10^{-3} s^{-1} up to that range for two single-phase materials, pure Ni and a solid solution of Ni-20wt.%Cu. Nickel is chosen due to its widespread use and engineering relevance, making it an ideal candidate for exploring material behavior across a broad range of strain rates and providing validation touchpoints against prior literature. Ni-20Cu is selected as a complementary material to introduce compositional and mechanical differences while ensuring comparable fabrication methods."

5) Page 3, line 72-74: Does this positive strain-rate dependence match other studies?

Authors' response: Yes, this is a typical signature of thermally activated plastic flow, where the positive strain-rate dependence is commonly observed in the quasistatic range and indicates a positive activation-volume mechanism of deformation. Furthermore, we point out in the discussion that the single value for nickel aligns with what is reported in the literature.

Changes to the manuscript: We have revised the sentence and included a reference to the literature.

"...The data in Figure 1 in this quasi-static range show a positive strain-rate dependence, which is a typical signature of thermally-activated plastic flow²², and the solid solution alloy is harder than the pure metal, also as expected."

6) Page 3, line 86: You cite papers that other methods can achieve strain rates of 10^4 - 10^6 , which is a region you have no data points for in the methods you did use, so why not also include experiments using such methods?

Authors' response: We mention in the manuscript that new instrumentation can achieve deformation rates up to 10^4 - 10^5 s^{-1} . To our knowledge, there is currently no available equipment capable of reaching strain rates of 10^6 s^{-1} . Yet, strain rates achieved by novel instruments are typically around 10^4 s^{-1} , with only a few exceptions reaching up to 10^5 s^{-1} , where the indentation depths involved are much lower than those

investigated in our study. Thus, including those methods would complicate the paper by introducing, e.g., size effects. One advantage of the current approach is that all the indentations are of similar scale.

No changes to the manuscript.

7) Figure 1: There is a huge variation in the (conventional) LIPIT results in terms of the harness calculated—what causes this extreme variation as compared to other results, and given the variance, how accurate is the slope of the line you have calculated from 10^6 - 10^8 /how do you know the trend you observe is indeed true? Again, the slope in this region seems to depend only on three data points, and one of those data points seems to be quite imprecise.

Authors' response: The observed variation can be attributed to several factors, including inherent experimental uncertainties due to minor deviations in impact velocities, as well as variations in the mass of impactors and indentation volumes, and the heterogeneity of the material response at ultra-high strain rates. It is important to emphasize that experiments at these strain rates are essentially challenging, and some degree of scatter is expected. However, we believe that variations observed in our results remain within an acceptable range for studies of this nature and still provide valuable insights into material behavior under extreme conditions. While there are three average data points plotted in the ultra-high strain rate regime, each of these averages is derived from a robust dataset comprising at least five and up to fifteen individual experimental tests. This dataset is considerably more extensive than what is typically reported in studies utilizing other low-throughput, high-strain-rate techniques. As such, we believe the presented data are reliable and represent meaningful trends.

No changes to the manuscript.

8) Page 5, line 132: You say you have a fixed hardness target, but you are also showing in this paper how hardness scales with strain rate, so what does “fixed hardness target” mean in this case? Also, what is the exact target used?

Figure 2a: Hard to see the different colors surrounding the lines and exactly what they correspond to, please clean up plot

Figure 2b: You have that the indentation depth for the cones is either 1 or 2 microns, but later say that the indentation depth shown in figure 4A is approximately 5 microns. Why include those cones in your plot if they aren't relevant experimentally because the indentation depths you are achieving are much greater?

Authors' response: We grouped together these comments as there seems to be a misunderstanding.

Fig. 2 is explanatory and purely “theoretical”, exploring the relationship between mass, velocity, and indentation depth under the assumption of a fixed hardness value of $H=3.5$ GPa. This value corresponds to the dynamic hardness of the material and serves as a baseline for illustrating how variations in impactor design and mass affect achievable strain rates and indentation depths. While hardness does scale with strain rate, a single representative hardness value was assumed here to isolate the effects of mass and velocity. The hardness could have taken any other value, the idea was to represent the experimental challenge in a figure.

Furthermore, in addition to fixing the hardness, one could also evaluate the velocity and mass required to achieve a specific indentation depth. This is particularly important since, for instance using nanoindentation, one could achieve a higher strain rate by keeping the indentation depth smaller; however, this substantially changes the size of the deformed area and could lead to changes in the material response due to sample size effects, or even reach the case in which only elastic interactions are involved.

In Figure 2, we selected two representative indentation depths (1 μm and 2 μm) to emphasize the challenges of lowering strain rates while keeping the size of the indentation constant: using LIPIT, reducing the impact velocity decreases both the indentation depth and the strain rate, irrespective of the specific material properties. The chosen depths also illustrate the limited working range for a given impactor design. Additionally, the plot shows that further lowering the velocity decreases the indentation depth up to the point of risks resulting in an elastic interaction between the impactor and the target. The same indentation depths were applied for other shapes to highlight these constraints.

We believe the main takeaway from Figure 2 is clear: reducing the strain rate generated by LIPIT for a given indentation depth requires maintaining a consistent indentation area while increasing the trailing mass to provide the necessary energy. Any shape that satisfies this condition—such as the cone or a small microfabricated cylinder with a large base—can effectively achieve this goal, as shown later in the manuscript.

Changes to the manuscript: We have introduced explanatory details prior to Fig. 2

“...The challenge is laid out in Figure 2A, which charts the relationship between mass and indentation depth for impactors of given velocities impacting on a target with a fixed hardness of $H=3.5$ GPa; the corresponding strain rate is displayed on the upper X-axis. This fixed hardness value represents the dynamic hardness of the target material and is used solely as a reference to simplify the analysis, although it might change somewhat with strain rate for many test materials. This figure includes alumina indenters of various shapes, and considers an empirical power law that establishes an upper limit of impact velocity as a function of impactor mass reported in Ref. ³³. This empirical power law represents the situation in which an increase in laser energy no longer results in a corresponding increase in velocity due to the activation of other mechanisms of energy dissipation, such as launch pad fracture. When using spherical particles, an order of magnitude decrease in strain rate requires an order of magnitude increase in particle size to achieve the same indentation depth (Figure 2A-blue region); by extension, the indentation area expands considerably (as shown in Figure 2B-blue region), moving the method beyond the micrometer scale. This illustrates the tradeoff inherent in reducing strain rates when using spherical particles: decreasing velocity and strain rate requires a very large increase in mass, which in turn increases the contact area and changes the scale of the test.”

9) Figure 2b: this does not need to be addressed in the paper, but I am curious as to what would happen if you launched a cylinder on the side—you would have the same amount of mass, but a much bigger footprint in your indentation. Maybe not as deep, but would you still achieve a greater volume of indentation and therefore have a different strain rate?

Authors' response: One could evaluate this situation in a manner similar to what was done in Figure 2. The hardness (H) can be expressed as the plastic work (W_p) over the indentation volume (V). For simplicity, let's assume that, if the mass of the impactor does not change when launching it on the side at the same velocity (v), the rebound velocity would be similar so that the plastic work remains the same in both situation (launching it straight (1) or on the side (2)). Again, for simplicity, let's approximate the indentation volume as the product of the indentation area (A) and the indentation depth (h). This gives:

$$H = \frac{W_p}{V} \sim \frac{W_p}{A_1 * h_1} \sim \frac{W_p}{A_2 * h_2}$$

which means:

$$h_2 \sim \frac{A_1}{A_2} h_1$$

And the strain rate in both situations would be:

$$\dot{\epsilon}_1 = \frac{v}{h_1}$$

$$\dot{\epsilon}_2 = \frac{v}{h_2} \sim \frac{v}{\frac{A_1}{A_2} h_1} \sim \dot{\epsilon}_1 \frac{A_2}{A_1}$$

Since $A_2 > A_1$, launching the cylinder on its side results in a higher strain rate (which will depend on the aspect ratio), but this comes at the expense of a reduced indentation depth ($h_2 < h_1$). This reduction in indentation depth is contrary to the goal of avoiding indentation size effects at lower strain rates.

No changes to the manuscript.

10) Figure 3a-c: How was the angle of the diamond selected? From puncture mechanics (see for example doi: 10.1109/ROBOT.2003.124185), the angle of impactor can have quite an effect on how the substrate responds

Authors' response: We agree with the reviewer that there are numerous studies, including those in nanoindentation, that demonstrate how the indenter geometry can influence the strain imposed and the resulting hardness values. In metals, these changes are generally rather small and not consequential to hardness measurement. Indeed, it is common practice to measure the “shape function” or “area function” of the indenter tip. Although this shape function is different across many experiments, they all give the same basic result; indifference to details of the shape is a cornerstone of hardness testing. Our “shape function” is reported in the manuscript: we employed sharp polished diamond indenters with an area function that can be approximated as equivalent to a sharp cone with an angle of about 115°. We use that exact same geometry across a range of strain rates from quasistatic to approximately 10^6 s^{-1} . Additionally, as highlighted in the response to reviewer 2, point 1, we compared results at strain rates around 10^6 s^{-1} using both sharp cone-like diamonds and two-stage-cylindrical-tip indenters. Our findings show no significant difference in the resulting hardness within this strain rate regime.

Changes to the manuscript: We have introduced a more precise discussion of the shape function of our tips in section ‘Methods-LIPIT impactors’

“...The final height was, in some cases, modified by laser-guided waterjet (Evolved Diamonds LLC, US) to tailor the impactor mass (Figures 2A-C). These sharp indenters have an area function matching a sharp cone with an angle of 115°.”

11) Figure 3e: Why are you using cylinders as your impactor shape when figure 2 states it is harder to achieve variable strain rates using cylinders? Why not also make these cones?

Authors' response: We refer the reviewer to our response provided to comments in point 8). This is clearly a minor misunderstanding. As discussed, reducing the strain rate generated by LIPIT for a given indentation depth requires maintaining a consistent indentation area while increasing the trailing mass to provide the necessary energy. Importantly, the indenter tips we fabricated are NOT simply cylinders, and thus not counterindicated by Fig. 2 as the reviewer suggests: they are two-stage geometries, where the cylinder tip is attached to a large trailing mass. This moves off the curve from Fig. 2 in a fundamental way.

Changes to the manuscript: We have gone through the paper to ensure that all references to the microfabricated tips avoid the use of the term “cylinder” or “cylindrical”, and emphasize their two-stage geometry instead. We believe this will avoid the confusion on this point.

12) Figure 4a and c: it may be nice to add a line to guide the reader to where the substrate is positioned in the camera frames

Authors' response: Thank you; we have included a reference of the substrate surface in Fig. 4a and c.

Changes to the manuscript: We have updated Fig. 4a and c, please refer to the main manuscript.

13) Page 8, line 187: Please put the achieved strain rates from the microfabricated diamonds

Authors' response: The corresponding strain rate for the microfabricated diamonds is already included in the manuscript. The relevant sentences are:

“Similarly, snapshots in Figures 4C show the impact event of a microfabricated diamond, containing a flat punch of $D \sim 35 \mu\text{m}$, impacting a Ni-20 wt.% Cu target at a velocity of 16 m s^{-1} . The impact resulted in an indentation of $\sim 5 \mu\text{m}$ depth (Figure 4D), representing an average strain rate of $3 \cdot 10^6 \text{ s}^{-1}$.”

Yet, we have extended our explanation before Fig. 4.

Changes to the manuscript: We have extended our explanation in the first paragraph following Fig. 3.

“...The impact resulted in an indentation of $\sim 5 \mu\text{m}$ depth (Figure 4D), representing an average strain rate of $3 \cdot 10^6 \text{ s}^{-1}$. The laser-diced polished diamonds, displayed in Figure 3A left, were launched at velocities comparable to those of the microfabricated diamonds, which permits us to directly establish that the details of the indenter shape are not critical to the measurement of hardness according to Eq. (1). The results align in terms of both the strain rate and resulting hardness for both nickel and nickel-copper specimens (see Supplementary Materials). The data points at strain rate of $\sim 10^6 \text{ s}^{-1}$ in Figure 1 thus include tests conducted with both microfabricated and laser-diced polished diamonds.”

14) Page 8, lines 186-195: When discussing the microfabricated diamonds, it is not clear if these experiments made it into figure 1. Are the two data points shown there from the two different diamond impactors? If so, please clarify which point corresponds to which impactor. Furthermore, could the microfabricated impactors be further tailored to achieve more strain rates? One of the weaknesses of this paper is the lack of data points in the higher strain rate values

Authors' response: Thanks for the comment. We have now clarified that point in the manuscript. Tests conducted with microfabricated diamonds are indeed included in Figure 1 (at around strain rates of 10^6 s^{-1}). Regarding tailoring the microfabricated impactors to achieve higher strain rates, it is indeed possible to adjust the mass of these impactors to extend the data range into higher strain rates. However, we believe the range already covered in the manuscript is one of the major achievements of this work, as no previous studies have presented multiple data points at these ultra-high strain rates. Conventional techniques, such as Kolsky bars, typically present only a few data points, and often at lower strain rates. Thus, we consider this a distinctive and valuable aspect of our study.

Changes to the manuscript: We have extended our explanation in the first paragraph following Fig. 3.

"...The impact resulted in an indentation of $\sim 5 \mu\text{m}$ depth (Figure 4D), representing an average strain rate of $3 \cdot 10^6 \text{ s}^{-1}$. The laser-diced polished diamonds, displayed in Figure 3A left, were launched at velocities comparable to those of the microfabricated diamonds, which permits us to directly establish that the details of the indenter shape are not critical to the measurement of hardness according to Eq. (1). The results align in terms of both the strain rate and resulting hardness for both nickel and nickel-copper specimens (see Supplementary Materials). The data points at strain rate of $\sim 10^6 \text{ s}^{-1}$ in Figure 1 thus include tests conducted with both microfabricated and laser-diced polished diamonds. "

15) Page 9, lines 199-200. It was very well done to use the same tips to do the lesser strain rate values. However, for the conventional LIPIT, the indenter was made of aluminum rather than diamond. 1. How will having different hardness indenters affect the measured hardness value? 2. Could the microfabricated diamonds have been microfabricated aluminum instead? Why not try to use the same material for all tests?

Authors' response: We would like to clarify that the indenters were made of diamond and alumina, not aluminum. Both are significantly harder materials than the alloys investigated, and we base our analysis on the assumption that all plastic deformation is confined to the substrate. Thus, there should be no difference in the measured hardness values when using either material. Regarding the suggestion to microfabricate alumina indenters instead of diamonds, while it is possible to use alumina in microfabrication, there is no particular advantage in doing so for this study. Diamond and alumina both serve as suitable materials for indenters due to their hardness, and the primary focus was on maintaining consistency in terms of indentation behavior across different strain rates, which we believe has been achieved.

Changes to the manuscript: We have added a sentence at the end of the section "Methods-LIPIT impactors".

"...For the case of diamonds, the mass of each impactor was measured using a piezoelectric crystal microbalance with a resolution of $1 \mu\text{g}$, while for alumina particles, the mass of each of them was calculated based on their measured diameters prior to experiment. In both cases, it was assumed that all plastic deformation was confined to the substrate. "

16) Page 9, line 202: Please add a citation to support this statement

Authors' response: We have added a reference to support the statement regarding when indentation size effects become important, which typically occurs at indentation depths below one micron (<https://doi.org/10.1016/j.jmrt.2024.06.071>).

Changes to the manuscript: We have added a reference to support our statement.

"...All of the experiments in Figure 1 also reflect measurements at the same size scale, with depths all in the range $2 - 15 \mu\text{m}$; this is well outside of the range where size effects are normally seen³⁵, and thus all the present data represent bulk measurements."

17) Page 9, line 212: You compare the strain rate sensitivity of Ni to those in literature, but what about Ni-20Cu? Why chose it as a substrate if there isn't data to compare it to?

Authors' response: Since this is similar to the reviewer's comment in point 4), we kindly refer to our response there.

No changes to the manuscript.

Reviewer #1 (Remarks to the Author):

The authors have answered all comments well. Congratulations.

Authors' response: We thank the Reviewer for the time and positive comments.

Reviewer #2 (Remarks to the Author):

While previous reviewer comments have been addressed to some extent, the concern about self-consistency of strain rate definition among different impactors remains inadequately resolved, that authors should resolve before publication:

The reviewers agree that the definition of hardness is self-consistent for conical and cylindrical diamond indenters. However, no such consistency has been demonstrated between the microfabricated impactors used for strain rates of 10^5 - 10^6 s⁻¹ and the alumina spheres used for strain rates of 10^8 s⁻¹. Since the model beyond the rate-sensitivity upturn (pages 10–11) relies on experimental results from both diamond and alumina impactors, and is highly sensitive to strain rate, it is crucial to establish a consistent strain rate definition across all impactors, regardless of their geometry or material. Without this, any attempt to develop a model would lack a solid experimental foundation.

Authors' response: We appreciate the Reviewer's comment. Given their shape differences, we expect a larger disparity between conical and cylindrical impactors than between conical and spherical ones, which are already compared. Unfortunately, comparing spherical and sharp impactors via LIPIT impacts in both hardness and strain rate is challenging, due to the absence of smaller sharp impactors equivalent to those used at lower strain rates. Nonetheless, to address the reviewer's concern, we have performed two additional tests following a similar approach to the one used for comparing conical and microfabricated diamond impactors:

(i) We have conducted impact tests at lower strain rates ($\sim 10^3$ s⁻¹) with a spherical diamond tip ($D=25$ μ m), at a similar depth of indentation. We then compared the results in terms of impact velocity, average strain rate and hardness, with those acquired using conical diamonds.

As observed in the following figure (A&C), the data generated using these two different indenter geometries are comparable (i.e., within uncertainty of one another) in terms of computed strain rate and resulting hardness.

Figure S1 (A) Hardness as a function of strain rate for pure 99.99% nickel containing data acquired by impact indentation employing a spherical indenter (green) and irregular particles launched by conventional LIPIT (purple). The inset includes a confocal image of an indent left by irregular particles, alongside an SEM image of the particles for reference. (B) Comparison of hardness values for spherical (\circ -blue) and irregular (\square -purple) alumina particles, impacting at similar velocities. (C) Comparison of hardness values for polished conical diamonds (\square -blue) and spherical (\circ -green) indenter, impacting at similar velocities.

(ii) Similarly, we have conducted LIPIT tests launching irregular alumina particles (see A&B) and compared those results with data from spherical alumina particles. These irregular particles produce impacts that are self-similar, more like a “sharp” indenter, and therefore are indeed more comparable to the other experiments. A confocal image of a sharp impact indentation is shown inset to Figure S1, and we note that it is very geometrically similar to what we obtain with large diamond indenters at lower strain rates. This approach therefore produced a robust test of indentation geometry in the extreme strain rate range of LIPIT.

Despite a higher spread of resulting hardness – likely due to uncertainties in estimating impactor mass – the associated strain rate and the average hardness with these sharp indenters remains comparable (within uncertainty) to the result with the spherical particles.

We therefore conclude that while impact experiments inherently involve a complex strain rate evolution, the strain rate definition used here, adopted from conventional nanoindentation, remains appropriate and captures the overall trends. These additional tests demonstrate that the

computed strain rate, and hardness values, are comparable across different impactor geometries and any discrepancies lead to no changes in our main conclusions.

Changes to the manuscript: These new data have been now added to the Supplementary Materials and reference to it has been included in the main manuscript, prior to Figure 4.

'The laser-diced polished diamonds, displayed in Figure 3A left, were launched at velocities comparable to those of the microfabricated diamonds, which permits us to directly establish that the details of the indenter shape are not critical to the measurement of hardness according to Eq. (1). The results align in terms of both the strain rate and resulting hardness (see Supplementary Materials). The data points at strain rate of $\sim 10^6$ s⁻¹ in Figure 1 thus include tests conducted with both microfabricated and laser-diced polished diamonds. Similarly, additional experiments have been conducted to validate the strain rate definition for spherical indenters used in conventional LIPIT tests, as detailed in the Supplementary Material.'

Reviewer #3 (Remarks to the Author):

The details provided in the reviewer's response have made the paper much stronger. I think this paper is worthy of publication, though I still feel that the strongest contribution is in the areas of LIPIT and showing the different geometric control, which does not perhaps merit publication in Nature Communications. Indeed, their claim that LIPIT has never been connected to mainstream testing methods (response to question 3, reviewer 3) is incorrect (for example, ACS Applied Materials Interfaces, 2023, 15, 32916-32935 for connection to experimental methods, ACS Macro Letters, 2024, 13, 3, 302-307 for connection to MD simulation). In addition, I still feel that the authors are relying on only a few data points to claim a universal curve over 11 orders of magnitude (Figure 2). While they addressed this point in their comments to the reviewers, I still believe, even with the experimental difficulty, they could fill in some of the areas, in particular in the 10⁰-10⁴ regime. I think to merit such bold claims, a few more data points in the intermediate regime are necessary. The authors have written a good paper, and I think it will have significance to the LIPIT and other high strain rate test community, but I do not think it rises to the level of universality they are claiming.

Authors' response: We thank the reviewer for the kind words and the supportive comment about our publication-worthiness. Clearly, we disagree with the judgment of the reviewer as to how much new data is needed to constitute a complete study. It is clear that more work would produce more data, and more data would seem more comprehensive, but this is a tautology. No paper would ever be "done" if every possible gap in any given dataset needed to be addressed. Science progresses in steps, and we remain confident that the step in this paper is a major one. It took years of effort to achieve this homogenization, and it has never been done before; we maintain that the advance is a significant one. Adding more datapoints would not change the trends we have mapped out, it would merely fill in a trend that we have already established here for the first time. As a case in point, we have now included new data with the spherical indenter (green points in Fig. S1) as requested by the second reviewer. These data are consistent with the trend established previously— adding new points is not changing the trends, it is merely adding points.

We hope that the reviewer appreciates the new data, and the broader point. It seems that the reviewer would not want to prevent the publication of important work on subjective grounds about how many extraneous data points are needed to support a trend, especially in the absence of technical deficiencies. Since we are the designers and authors of the work, we request that our judgment stand.

The new datapoints (green) from Fig. S1 were added in the middle of the strain rate regime of concern to the reviewer.

Self-consistent hardness measurements spanning eleven decades of strain rate on a single material surface

General questions/comments:

- Overall, this is a nice paper with a very sound premise: try to use one unifying method to assess the materials properties across a large range of strain rates, with a particular interest in how to achieve moderately to ultra-high strain rates. It nicely blends different techniques into one equivalent mathematical formula to connect different results
- This manuscript claims to study hardness over 11 orders of magnitude. While technically correct, this statement is somewhat misleading. It characterizes the quasi-static regime, 10^{-3} - 10^0 , then has a single data point at 10^3 , then two data points at 10^6 , then another at 10^8 . This means that over 6 orders of magnitude, there is a single data point, which means interpolation across this regime is highly dependent on that single data point. Furthermore, the high strain rate regimes are also a series of three points. It is not that I do not believe these points or these trends, but I do not think there is enough data to fully support their claims across these regions
- All discussion is lifted from previous papers: validating and perhaps refining slightly what has been seen already with one test, but not adding anything new in terms of understanding what is causing the hardness
- In my opinion, the most novel part of the paper is a methodological development for LIPIT, namely, that using different shaped impactors can control the strain rate, is an important finding for the LIPIT community, but does not have obvious significance for others outside. In the world of puncture mechanics, such in studying needles, the shape of an indenter, albeit at lower strain rate than LIPIT, has been known to affect the response of a material. So this is a useful finding, but not necessarily novel. Thus, I believe this paper merits publication, but not necessarily in Nature Communications, and not without significant revision

Some specific questions/comments

- Why were Ni and Ni-20Cu chosen for this study?
- Page 3, line 72-74: Does this positive strain-rate dependence match other studies?
- Page 3, line 86: You cite papers that other methods can achieve strain rates of 10^4 - 10^6 , which is a region you have no data points for in the methods you did use, so why not also include experiments using such methods?

- Figure 1: There is a huge variation in the (conventional) LIPIT results in terms of the harness calculated—what causes this extreme variation as compared to other results, and given the variance, how accurate is the slope of the line you have calculated from 10^6 - 10^8 /how do you know the trend you observe is indeed true? Again, the slope in this region seems to depend only on three data points, and one of those data points seems to be quite imprecise.
- Page 5, line 132: You say you have a fixed hardness target, but you are also showing in this paper how hardness scales with strain rate, so what does “fixed hardness target” mean in this case? Also, what is the exact target used?
- Figure 2a: Hard to see the different colors surrounding the lines and exactly what they correspond to, please clean up plot
- Figure 2b: You have that the indentation depth for the cones is either 1 or 2 microns, but later say that the indentation depth shown in figure 4A is approximately 5 microns. Why include those cones in your plot if they aren’t relevant experimentally because the indentation depths you are achieving are much greater?
- Figure 2b: this does not need to be addressed in the paper, but I am curious as to what would happen if you launched a cylinder on the side—you would have the same amount of mass, but a much bigger footprint in your indentation. Maybe not as deep, but would you still achieve a greater volume of indentation and therefore have a different strain rate?
- Figure 3a-c: How was the angle of the diamond selected? From puncture mechanics (see for example doi: 10.1109/ROBOT.2003.124185), the angle of impactor can have quite an effect on how the substrate responds
- Figure 3e: Why are you using cylinders as your impactor shape when figure 2 states it is harder to achieve variable strain rates using cylinders? Why not also make these cones?
- Figure 4a and c: it may be nice to add a line to guide the reader to where the substrate is positioned in the camera frames
- Page 8, line 187: Please put the achieved strain rates from the microfabricated diamonds.
- Page 8, lines 186-195: When discussing the microfabricated diamonds, it is not clear if these experiments made it into figure 1. Are the two data points shown there from the two different diamond impactors? If so, please clarify which point corresponds to which impactor. Furthermore, could the microfabricated impactors be further tailored to achieve more strain rates? One of the weaknesses of this paper is the lack of data points in the higher strain rate values
- Page 9, lines 199-200. It was very well done to use the same tips to do the lesser strain rate values. However, for the conventional LIPIT, the indenter was made of

aluminum rather than diamond. 1. How will having different hardness indenters affect the measured hardness value? 2. Could the microfabricated diamonds have been microfabricated aluminum instead? Why not try to use the same material for all tests?

- Page 9, line 202: Please add a citation to support this statement
- Page 9, line 212: You compare the strain rate sensitivity of Ni to those in literature, but what about Ni-20Cu? Why chose it as a substrate if there isn't data to compare it to?